# Feature Selection Strategies for Deep Learning-Based Classification in Ultra-High-Dimensional Genomic Data

**DOI:** 10.3390/ijms26167961

**Published:** 2025-08-18

**Authors:** Krzysztof Kotlarz, Dawid Słomian, Weronika Zawadzka, Joanna Szyda

**Affiliations:** 1Biostatistics Group, Department of Genetics, Wroclaw University of Environmental and Life Sciences, 51-631 Wroclaw, Poland; krzysztof.kotlarz@upwr.edu.pl (K.K.); 130732@student.upwr.edu.pl (W.Z.); 2National Research Institute of Animal Production, 32-083 Balice, Poland; dawid.slomian@iz.edu.pl

**Keywords:** dimensionality reduction, deep learning, feature selection algorithms, mixed linear model, multi-class classification, SNP, whole-genome sequencing

## Abstract

The advancement of high-throughput sequencing has revolutionised genomic research by generating large amounts of data. However, Whole-Genome Sequencing is associated with a statistical challenge known as the p >> n problem. We classified 1825 individuals into five breeds based on 11,915,233 SNPs. First, three feature selection algorithms were applied: SNP-tagging and two approaches based on supervised rank aggregation, followed by either one-dimensional (1D-SRA) or multidimensional (MD-SRA) feature clustering. Individuals were then classified into breeds using a deep learning classifier composed of Convolutional Neural Networks. SNPs selected by SNP-tagging yielded the least satisfactory F1-score (86.87%); however, this approach offered rapid computing time. The 1D-SRA was less suitable for ultra-high-dimensional data due to computational, memory, and storage limitations. However, the SNP set selected by this algorithm provided the best classification quality (96.81%). MD-SRA provided a good balance between classification quality (95.12%) and computational efficiency (17x lower analysis time, 14x lower data storage). Unlike SNP-tagging, SRA-based approaches are universal and are not limited to genomic data. This study addressed the demand for efficient computational and statistical tools for feature selection in high-dimensional genomic data. The results demonstrate that the proposed MD-SRA is suitable for the classification of high-dimensional data.

## 1. Introduction

Over the past several decades, there has been a significant increase in the availability of genomic data for a rapidly increasing number of individuals. Due to the decreasing cost of using high-throughput sequencing methods, researchers have been able to examine genomes with DNA sequence accuracy [1]. However, the increased availability of genomic data is accompanied by increasing challenges associated with storage, bioinformatic processing, and statistical analysis. Storage not only requires large hard disc spaces but is also associated with substantial data loading time, as well as writing intermediate and final result files, activities that are very difficult or often impossible to parallelise and execute on a GPU. Parallelisation of data processing on a CPU (using vectorised code, applying OpenMP, or even MPI directives) or a GPU is typically possible for bioinformatic data processing and thus enables faster computations. However, often limited factors include the number of available parallel processing units and RAM, especially if data cannot be analysed on HPC architectures owing to access or privacy constraints. Bioinformatic processing of such data is especially challenging due to its underlying heterogeneity, which includes various genomic-related omics that are not always all available for each subject, while statistical inferences may be difficult because of missing data patterns that are not always MAR (missing at random) and, in the particular context of classification, by a severe class imbalance. Therefore, the application of machine learning, including deep learning (DL), approaches that provide flexibility in handling the above-mentioned characteristics, has been gaining importance recently [2].

The main focus of our study was on statistical problems related to high-dimensional genomic data processing [3,4,5]. They comprise the following:
Problems with the accurate estimation of model parameters. More specifically, as demonstrated by Giraud [3], standard errors of estimates $E{\left(\hat{\beta }-\beta \right)}^{2}$ linearly increase with an increasing number of model dimensions that, as a consequence, make statistical and biological inferences based on $\hat{\beta }$ less reliable. Furthermore, when the variation of a dependent variable is described by numerous independent covariates (features), false positive associations can arise due to fitting patterns to “noise” [5]. The accurate estimation of model parameters in a high-dimensional setup is also hampered by numerical inaccuracies that may result from many numerical operations, often on the verge of overflow or underflow.Interpretability of model parameters due to correlations between subsets of features. This induces the risk of fitting false patterns of the variation of the dependent variable to the noise in the independent variables.The applicability of traditional hypothesis testing, based on *p*-values for which the multiple testing correction may fail due to the violation of the underlying assumption of independence between single tests, leading to inflated Type I error rates.Especially for the specific task of classification, in a multidimensional space, many data points lie near the true class boundaries, leading to ambiguous class assignments.

In reality, especially in the case of whole-genome sequence (WGS) data, the underlying true dimensionality of the model is not as complex as the number of available features. Therefore, feature selection (FS) is not only essential to circumvent the challenges mentioned above, but also helps identify biologically relevant features for downstream analysis [6]. There are many existing FS techniques that are widely classified into basic, hybrid, and ensemble approaches [7]. To our knowledge, no formal comparison addressing their computational efficiency, coupled with classification quality, exists for WGS data. Our study focused on an ensemble approach designed for ultra-high-dimensional data. Ensemble approaches combine several models created from the same dataset to provide the best accuracy and robustness of the feature selection process [8]. To avoid the above-mentioned problem with *p* values, the final selection of relevant features is based on a cutoff that is either arbitrarily defined or estimated from the data. A key step in ensemble-based feature selection is rank aggregation (RA), which combines feature importance scores from many models. RA creates an overall rating of features by combining the internal ranking of features within reduced models characterised by different model performances. Feature ranking within a particular reduced model can be based on feature estimates, for example, whereas the performance of reduced models is typically expressed by the reduced model fit quality. The final RA, which is often the most challenging step both statistically and computationally, can imply a range of aggregation strategies. For instance, a linear mixed model (LMM) can be used because of its robustness to the violation of asymptotic properties [9], such as normality or homoscedasticity, its ability to handle complex data structures by the possibility of incorporating feature correlation information directly into the model, and by elegant handling of p >> n problem through imposing a $N(\mu ,\sigma^{2})$ shrinkage on model estimates [10]. Furthermore, in our analysis, the problem of data storage was addressed by implementing memory mapping, which allowed us to avoid holding the entire dataset in memory, as well as by CPU- and GPU-based task parallelisation and vectorisation, which were directly implemented in the rank aggregation procedure and DL classification.

The major goal of the study was to compare three feature selection algorithms of varying computational and statistical complexity, to define an optimal subset of features for the deep learning-based multi-class classification in a ultra-high-dimensional setup of strongly correlated features. For this purpose, WGS data from 5063 bulls characterised by 33,595,340 single nucleotide polymorphisms (SNPs) was used to illustrate the three feature selection algorithms. The objective of this study was to provide an SNP subset for the classification of bulls into five classes represented by breeds (Angus, Brown Swiss, Holstein, Jersey, and Norwegian Red). The FS implemented a purely mechanistic approach to reduce the correlation between SNPs expressed by Linkage Disequilibrium (LD) (*SNP tagging*) and two approaches considering statistical contexts by assessing the importance of a particular SNP for the fit of a classification model by applying a multinomial logistic regression model followed either by one-dimensional feature clustering (*1D-SRA*) or by multidimensional feature clustering (*MD-SRA*).

The outline of the publication was as follows: (i) we described the methodology of SNP selection and breed classification, (ii) we explored the differences in sets of selected SNPs and the computational efficiency of the applied methods, and (iii) we moved to the actual multi-class classification via DL architectures, using the three SNP sets identified in part (i) as input features, while in the last section (iv), we concluded the study with a discussion of major differences between the FS approaches, recommendation of use, and drew directions for future research in this area.

## 2. Results

### 2.1. Feature Selection

In this section, we present the results of SNP pre-selection using the three proposed feature selection algorithms: (i) SNP tagging based on LD, (ii) 1D-SRA implementing a linear mixed model in the aggregation step, and (iii) MD-SRA implementing aggregation by weighted multidimensional clustering. The computational efficiency of these SNP selection algorithms varied considerably, reflecting the complexity of the analysis associated with each approach (Table 1).

***SNP tagging*:** The LD pruning method, which is the simplest in terms of computational demand, was used as the baseline for our comparison. It took 74 min to complete. The initial set of 11,915,233 SNPs was reduced to 773,069 SNPs (a reduction rate of 93.51%). The disc storage requirements were very small since this approach retained only the pruned set of SNPs, without intermediate files.

***The 1D-SRA*:** This approach involved fitting multinomial logistic regression models followed by rank aggregation based on LMM. Since many reduced logistic regression models were fitted, this approach required considerable CPU resources. Further, CPU time was needed to compute a design matrix Z for LMM and obtain LMM solutions. In addition, the procedure required storing and reading multiple files on a hard disc without the possibility of completing the whole 1D-SRA pipeline in memory. The wall clock time of 2790 min (46 h and 30 min) was 37.7 times longer than that for LD pruning. The initial set of SNPs was reduced to 4,392,322 SNPs, achieving a reduction rate of 63.14%. Since this approach required substantial hard disc storage of estimates of the logistic regression model effects needed for the aggregation step by the LMM, it resulted in terabytes of file size (3.1 TB for the dataset analysed in this study); moreover, the Z matrix was also stored on a disc by implementing a memory mapping approach and then rewritten into the plain text file required by the mix99 software.

***The MD-SRA*:** This approach provides the advantages of incorporating statistical benefits and retaining computational efficiency. The wall clock time of 2 h and 40 min was only 2.2 times longer than that of the LD-based approach. The original set of SNPs was reduced to 3,886,351 SNPs, providing a reduction rate of 67.39%. The hard disc space required for the reduced feature performance matrix (227 MB for the dataset analysed in this study) was markedly lower than in the *1D-SRA* method because of fewer intermediate steps.

Both the *1D-SRA* and *MD-SRA* approaches selected a very similar number of SNPs marking features relevant for classification. However, the overlap in SNPs between the two approaches was very low. Interestingly, the *SNP tagging* approach shared an equivalent overlap with *1D-SRA* and *MD-SRA* (Figure 1).

Figure 2 visualises examples of features selected for two genomic regions of chromosome 14. A 100 kbp region (14:554,070–14:662,687) that harbours the DGAT1 gene, representing the genomic region under strong artificial selection in dairy cattle and thus characterised by very strong LD. A region of 100 kbp (14:80,000,175–14:80,099,518) is located on the distal part of the chromosome and represents a region without coding sequences (i.e., with low LD).

### 2.2. Classification

As summarised in Table 2, all three feature selection algorithms resulted in high F1-scores that indicated a good balance between precision and recall. The high AUC suggested a good ability to separate the classes. For the validation datasets, the highest AUC (averaged across the validation datasets) of 0.992 was achieved for *1D-SRA*, closely followed by *MD-SRA*, with AUC = 0.984. *SNP tagging* resulted in the lowest, albeit still high, AUC of 0.963. The same ranking of feature selection approaches was obtained, considering the F1-score metric with *1D-SRA*, *MD-SRA*, and *SNP tagging* scoring of 90.09%, 88.22%, and 74.65%, respectively. Apart from a very good classification performance of all three approaches in the validation datasets, considerable differences in their robustness, expressed by a standard deviation of the metrics, were observed. The *1D-SRA* was the most robust, with the lowest AUC and F1-score standard deviations. The *MD-SRA* was intermediate, with a standard deviation of AUC approximately three times higher and a standard deviation of the F1-score twice as high as the *1D-SRA*. The classification quality of *SNP tagging* was the least robust, with a standard deviation of the AUC being six times higher and a standard deviation of the F1-score being three times higher than for *1D-SRA*. Regarding the classification quality of the test dataset expressed by the F1-score, a very similar situation to that for the validation test was observed. The *1D-SRA* and *MD-SRA* provided a very similar classification quality of over 95%, while *SNP tagging* resulted in a good but lower F1-score of 86.87%. The AUC was very similar across all three approaches, with *1D-SRA* and *MD-SRA* achieving AUCs of 0.997 and 0.998, respectively, while *SNP tagging* resulted in a slightly lower AUC of 0.985. The confusion matrix for the test dataset across individuals is shown in Figure 3.

## 3. Discussion

Our study explored three procedures of feature pre-selection in ultra-high-dimensional data in the context of classification, as illustrated by the classification of bulls into five breeds based on their SNP genotypes identified from WGS. The procedures ranged from a mechanistic approach based solely on exploring pairwise LD between SNPs to computationally and statistically complex approaches that incorporate biological information. The proposed methods consider different sources of information contained in a dataset, each with its own strengths and limitations. *SNP tagging* is a well-known approach highly used in genomics, as it was explicitly designed to prune polymorphic genetic variants, such as SNPs, before further downstream analyses [10]. Since this method is limited to this specific data structure (SNPs), its high computing efficiency makes it a reasonable option for feature selection, albeit only in genomics. An obvious disadvantage is that it cannot be applied to other types of highly dimensional data. The *1D-SRA* is a general approach [7]. It has a wide range of applications that extend beyond biology. However, this increased flexibility comes with a higher data processing cost and computational complexity, which increases with increasing dimensionality of the datasets. Due to its complexity, maximal handlebar dimensions are restricted by available computational resources, including not only memory, but also hard disc space, time, and software. The latter strongly depends on the choice of the aggregation model. In the original application [7], aggregation via penalised regression or random forest was applied, while we implemented LMM. The advantages of using LMM lie not only in its robustness toward the violation of theoretical model assumptions, for example, of data normality [11], but mostly in its ability to handle correlated data with a complex covariance structure (although the latter was not required in our study). The ability of LMMs to include both the feature and the residual covariance directly into the model prevents false positive associations that, in the genomic context, may arise from ascertainment bias due to population structure or varying degrees of family relationship [11]. *MD-SRA* is a modified approach proposed by Jain and Xu [6], who proposed de novo in our study to avoid the problems associated with the aggregation model. It aims to balance the advantages of the first two approaches, as it applies to high-dimensional datasets not only from the area of biology, but also allows for efficient computing performance and low computing resources usage. It allows the capturing of interactions between features without the need to fit the final aggregation model, which is the most computationally intensive element of 1D-SRA. Still, it achieves a very similar classification quality with markedly reduced computation times compared to 1D-SRA. In our application, *MD-SRA* demonstrated 17 times shorter analysis time and 14 times shorter data storage. Due to its good classification performance coupled with high computational efficiency, *MD-SRA* appears to be a desirable feature selection algorithm for the classification of ultra-high-dimensional data.

Nowadays, in machine learning, with the advancement of large datasets with millions of records and explanatory variables, feature pre-selection aims not only to identify relevant features but also to reduce noise that poses the danger of fitting fake patterns to data that are not functionally relevant and that impede the prediction quality of new data, non-seen by the trained model. Furthermore, given the rapidly growing sizes of available datasets, it is crucial to develop computationally efficient algorithms [12]. Conventional machine learning methods, such as penalised regression (e.g., LASSO), offer simultaneous feature pre-selection and feature importance estimation by selecting features that significantly contribute to prediction or classification [6]. However, these methods may not always capture interactions between features in high-dimensional data. Moreover, in high-dimensional data, feature selection that is solely based on statistical significance expressed by *p*-values may no longer be relevant since the existing multiple testing correction approaches neither scale with the very high multiple testing dimensionality nor consider the correlation between tests (e.g., the Bonferroni correction) [13]. Therefore, the grouping of features as relevant or irrelevant for downstream analysis is currently a widely used approach [14]. Furthermore, our study used a dynamic threshold to differentiate between the relevant and irrelevant features. This approach was demonstrated by Jain and Xu [6], who implemented it as 1D K-means clustering and later extended it to multidimensional K-means clustering, allowing the classifier to incorporate various model quality metrics. The algorithm proposed in our study can identify and prioritise features by dynamically adjusting the relevant feature selection threshold in a multidimensional space defined by feature estimates from multiple reduced models. Such a low-complexity approach as K-means clustering performs accurately and efficiently in multiple dimensions, especially in multiple dimensions, and the identification of clusters corresponding to the relevant features is not straightforward [15].

A significant challenge in feature selection, as highlighted in our study and many others [16,17], is that the ranking of features is based on model fit and not model prediction quality, so feature selection prioritises features that make the model fit the training data well. However, this does not always lead to better predictions of new data. Also, in our study, *1D-SRA* and *MD-SRA* used model fit (expressed as $MP_{k}$) for ranking/weighting of features, while in *SNP tagging*, model fit was not considered. Although SRA-based algorithms do not select the same features (see Figure 2), their classification quality was still very similar. This similarity appears because, in classification, it is less important which of the highly correlated features is selected. Although not overlapping, the features chosen by the two algorithms are located close to each other. Furthermore, regions with high Linkage Disequilibrium (LD) (Figure 2A), which correspond to sets of highly correlated SNPs, could be characterised by markedly fewer SNPs than regions with low SNP correlation (Figure 2B). However, when the classification performance of the CNN classifier was evaluated based on the features preselected by the three algorithms, *1D-SRA* was highlighted as the best method in terms of classification quality and robustness. It demonstrated the highest Macro F1-score, the highest stability of the classification quality metrics across cross-validation datasets, and the best class prediction performance for the test dataset. Very similar, but slightly lower, classification metrics were obtained using the *MD-SRA*. However, the high demands of *1D-SRA* on computational resources limit its application for feature selection in ultra-high-dimensional setups. These favour *MD-SRA* for the practical application of feature selection in the case of ultra-high-dimensional data. SNP tagging produced the least satisfactory classification quality metrics, but still resulted in good performance.

Since the advent of ever-growing dimensions of datasets, feature selection is and will be an important data analysis step. Future research applications regarding the algorithm proposed in our study would explore ways to differentiate between important and unimportant features after ranking and evaluating the performance of the three approaches to predict quantitative outcomes. Specifically, although K-means clustering worked well in our investigation, it is worthwhile to investigate more complex feature clustering techniques such as hierarchical clustering, Gaussian Mixture Models, or other advanced methods available. However, these approaches have disadvantages, particularly with regard to computational limitations. Advanced clustering methods often require significantly more computational resources, particularly when dealing with large datasets, leading to longer processing times, which may not be feasible in real-time applications [18,19]. It is worth considering that some clustering approaches perform better in one-dimensional data than in multidimensional spaces [20]. Regarding prediction, it is important (i) to differentiate between the quality of prediction for new data ascertained from a similar or from a different general population, to assess which approach is more robust towards potential ascertainment bias, and (ii) to explore the prediction quality for data that is dynamic across an additional dimension, such as, e.g., populations undergoing strong natural or artificial selection pressures across time.

## 4. Materials and Methods

### 4.1. Sequenced Individuals and Their Genomic Information

The 1000 Bull Genomes Project Run 9 consists of 5063 whole-genome sequences of Bos taurus bulls with 33,595,340 biallelic SNP genotypes called by GATK software [21] and imputed and phased by Beagle 4.0 [22]. Coverage across these sequences ranged from 1.97x to 173.10x. The dataset included 140 breeds. The subset of this dataset analysed in our study (Figure 4) consisted of individuals with international IDs, characterised by a genome-averaged coverage of at least 8, and belonging to breeds represented by at least 150 individuals. SNP preprocessing was carried out with bcftools 1.10.2 [23] in parallel mode using 25 threads and involved removing polymorphisms with a call rate less than 95%, a minor allele frequency below 5%, and multiallelic SNPs. After filtering, 1825 individuals from five breeds (Angus, Brown Swiss, Holstein, Jersey, and Norwegian Red) were included in the downstream analysis, resulting in 11,915,233 SNPs per individual. This dataset was divided into a training dataset of 1460 individuals and a test dataset of 365 individuals. Splitting was performed to obtain a balanced set of breeds between the training and test datasets.

### 4.2. Feature Selection

The subset of SNPs and animals defined by the training dataset was used to select features, i.e., SNPs, for the DL-based classifier. The algorithms used for the feature selection are described below.

#### 4.2.1. SNP Aggregation Based on Linkage Disequilibrium (SNP Tagging)

This approach was based solely on reducing the correlation between SNPs that arise due to local Linkage Disequilibrium (LD). PLINK 1.9 [24] was used to select SNPs (tagSNPs) representative of a given genomic region. The LD between pairs of SNPs was quantified using the R^2^ metric, and tagSNPs were selected for genomic regions spanning 100,000 bp, considering a moving window of 1000 bp. The maximum LD threshold for determining a tagSNP was $R^{2}\;$= 0.5.

#### 4.2.2. SNP Selection-Based on One-Dimensional Supervised Rank Aggregation (1D-SRA)

The supervised rank aggregation (SRA) methodology implements supervised learning and rank aggregation for feature selection. It utilises a statistical context of SNPs by quantifying their importance for classification by fitting a multinomial logistic regression model to breed classes. The algorithm involved the following steps (Algorithm 1, Figure 5A):From a dataset comprising N = 1825 individuals and P = 11,915,233 SNPs, K = 47,660 reduced datasets were sampled, each containing all individuals but a random subset of S = 250 SNPs.For each of the reduced datasets, K multinomial logistic regression models were fitted for B = 5 breed classification (Equation (1)):(1)$$\log\left(\frac{P_{b}\left(x\right)}{P_{B}\left(x\right)}\right)=\alpha +\beta X,$$
where $P_{b}$ represents the estimated breed classification probability encoded as an N×B incidence matrix, α is a 1×B vector of breed-specific intercepts, β is a B×S matrix of breed-specific SNP estimates with the corresponding design matrix X containing SNP genotypes coded as follows: 0/0 represented as 0, 0/1 represented as 1, and 1/1 represented as 2.Model performance of each of the K models ($MP_{k}$) was expressed by the quality of fit of each of the multinomial logistic regression models quantified by the cross-entropy loss ($CE_{k}$) metric (Equation (2)):(2)$${MP}_{k}=CE_{k}=-\frac{1}{N}\overset{N}{\underset{i=1}{\sum }}\overset{B}{\underset{b=1}{\sum }}\left[y_{ib}\times log(p_{ib})\right],$$
where $y_{ib}$ represents the true classification of individual i to breed b and $p_{ib}$ is the corresponding probability estimated by model (1), which is given either by $p_{ib=B}=\frac{1}{1+\sum_{b=1}^{B-1}{exp(\alpha }_{b}+\beta_{b}x_{i})}$, for the reference breed (B) or, for each non-reference breed (b), by $p_{ib}=\frac{{exp(\alpha }_{b}+\beta_{b}x_{i})}{1+\sum_{b=1}^{B-1}{exp(\alpha }_{b}+\beta_{b}x_{i})}$.The resulting model performance matrix C is a sparse matrix of P columns containing feature performance (FPs) and ${MP}_{k}$ in the P+1 column. K rows corresponding to each reduced model. Precisely, each row contains FPs of S SNPs fitted in the *k*-th reduced model, expressed by $max(\hat{\beta })$ for each SNP, while FPs of the remaining P − S SNPs were set to zero, and the quality of fit of this model was expressed by the reciprocal of the cross-entropy loss $\frac{1}{{MP}_{k}}$. Due to the large dimensions of C, its disc storage was resolved using the memory map approach implemented in the NumPy library (version 1.24.3) [25].The aggregation step combined the effects of all SNPs from multinomial logistic regression reduced models using the following mixed linear model:(3)MP= μ+Za +ε,
where MP [K×1] represents the vector of fit of multi-class regression reduced models expressed by the reciprocal of cross-entropy loss $\frac{1}{{MP}_{k}}$, μ is a general mean, a [P×1] is a vector of random SNP effects with a pre-imposed normal distribution $N(0,I{\delta_{a}}^{2})$, with the corresponding design matrix ***Z*** [K×P] containing $max(\hat{\beta })$ of SNP effects resulting from the reduced models and is thus equivalent to the first *P* columns of the model performance matrix C**,** and **ε** [K×1] is a vector of random residuals distributed as $N(0,I{\delta_{\varepsilon }}^{2})$. Model parameters were estimated using Mix99 software [26], assuming the variances of ${\delta_{a}}^{2}=\frac{0.3}{P}\;{{\cdot \delta }_{y}}^{2}=6.5\cdot 10^{-5}$ and ${\delta_{\varepsilon }}^{2}=\frac{0.7}{P}{{\cdot \delta }_{y}}^{2}=1.5\cdot 10^{-4}$ as known (i.e., not estimated) because in such a large system of dense equations ${\delta_{a}}^{2}$ and ${\delta_{\varepsilon }}^{2}$ would have been technically impossible to estimate. Moreover, the goal of fitting the model is not to estimate SNP effects but to provide a ranking of features that will further be used for DL-based classification. Due to the large dimensions of the equation system, a direct solution by solving the mixed model equations was not possible numerically and computationally. Therefore, the iterative preconditioned conjugate gradient (PCG) method was applied, in which the preconditioned matrix was defined based on block diagonal elements of the coefficient matrix of the mixed model equations [27]. The CD convergence criterion of $1.0\cdot 10^{-13}$ was set as the stopping criterion for PCG, and the solver was run in parallel on 40 CPUs.The final step in the feature selection process comprised grouping the SNP effect estimates ($\hat{a}$) into sets *relevant* and *irrelevant* for classification, so that the relevant features were further used to train the DL-based classifier. This was performed by applying the one-dimensional K-means classification algorithm to define two clusters, implemented in the Scikit-learn library (version 1.3.0) [28]. Using K-means allowed us to avoid setting up an arbitrary cutoff value to define relevant and irrelevant SNP sets. This procedure assumed that features with greater importance for classification resulted in models with high MP, which corresponds to low cross-entropy loss [7].

The following pseudocode summarises the *1D-SRA* algorithm. Note that the code was optimised for clarity of presentation and not for computational efficiency.
**Algorithm 1:** 1D-SRA**input:**   Feature data X (N x P)      Target class Y (N x 1) **constants:** P = 11,915,233 // total number of features (SNPs)      S = 250 // reduced number of features      K = 47,660 // number of reduced models **step A1**: generate reduced data sets and assess performance of reduced models shuffle the vector of all P features **for** each reduced model k in K // over reduced models  randomly sample S features without replacement from the   shuffled vector   fit a multinomial logistic regression model to N individuals   described by S features (dependent variable–target class   independent variable–features)  collect model_performance_k = cross-entropy loss   **for** each s in S // over features     feature_performance_k_s = max(SNP effect estimate)     feature_index_k_s = feature Id from feature vector   **end for**
**end for**
**step A2**: generate model performance matrix C generate a model performance matrix of K rows and P + 1 columns filled with zeros **for** each k in K // over reduced models   C[k,P + 1] = model_performance_k   **for** each s in S // over features     C[k,feature_index_k_s] = feature_performance_k_s   **end for**
**end for**
**step A3**: supervised rank aggregation based on matrix C run LMM (dependent variable–model performance C[:,P + 1], independent variable–feature performance C[:,1:P] collect 1:P vector of LMM parameter estimates **step A4**: feature selection based on LMM parameter estimates define 2 clusters of LMM parameter estimates based on 1D-K-means clustering

#### 4.2.3. SNP Selection Based on Multidimensional Supervised Rank Aggregation (MD-SRA)

Because of the very high dimensions of the equation system underlying LMM, which was used for feature aggregation in 1D-SRA and was computationally demanding, an alternative, less computationally intensive approach was proposed. In this approach, the aggregation of features by LMM was substituted by the direct clustering of multiple model performance matrices C with multidimensional K-means. The algorithm comprised the following steps (Algorithm 2, Figure 5B):Five model performance matrices C1−C5 were created by repeating steps 1–4 described above. Since the performance of a single feature occurs only once within each model performance matrix, K rows of each model performance matrix were collapsed to form one dense vector of non-zero FPs, forming V1−V5 model performance vectors.*Relevant* and *irrelevant* groups of features were defined by assuming two clusters in five-dimensional K-means clustering of matrix ***V*** that was formed by appending dense vectors V1−V5. Matrix ***V*** was weighted by the multiplication of each $FP_{ij}$ by the reciprocal of cross-entropy loss $\frac{1}{{MP}_{k}}$ of the corresponding reduced model. The L-2 norm of the five-dimensional coordinates of the cluster centroids was used to define the cluster containing the relevant features, which corresponded to the higher norm in five-dimensional space.

The following pseudocode summarises the *MD-SRA* algorithm.
**Algorithm 2:** MD-SRA**input:**   Feature data X (N x P)      Target class Y (N x 1) **constants:** P = 11,915,233 // total number of features (SNPs)      S = 250   // reduced number of features      K = 47,660  // number of reduced models      W = 5    // number of model performance               vectors **step B1**: generate dense model performance matrix V **for** each v in W // over model performance vectors   execute step A1 of 1D-SRA algorithm   execute step A2 of 1D-SRA algorithm   transform performance matrix C by applying C[k,1:P] · C[k,P + 1]   v[1:P] = non-zero feature performance from each P of matrix C   generate matrix V by appending dense vectors v**end for****step B2**: feature selection based on weighted elements of V define 2 clusters of matrix V based on MD-K-means clustering

**Figure 5 ijms-26-07961-f005:**
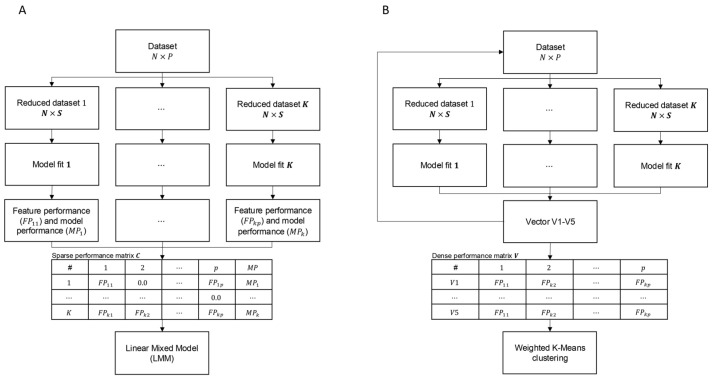
Representation of feature selection algorithms (**A**) *1D-SRA* and (**B**) *MD-SRA*.

### 4.3. Classification

The classification was based on the DL and was implemented in Python 3.13.1 via the Keras interface [29] to the TensorFlow library v.2.14.0 [30]. The following architecture was implemented as follows (Figure 6):The input consisted of the SNP genotypes that resulted from each of the three feature selection algorithms, comprising tagSNPs selected by *SNP tagging*, or SNPs representing the relevant cluster selected by *1D-SRA* or *MD-SRA*, respectively, as well as the breed class assignments for each individual (five breeds).First, in the convolution phase, (i) a sequential 1D Convolutional Neural Network (1D-CNN) was applied, followed by the Rectified Linear Unit (ReLU) activation function $f_{ReLU}(z_{i})=max(0,z_{i}),$ (ii) a batch normalisation (BN) layer, and (iii) a max pooling layer with a pool size of five. In particular, for each 1D convolutional layer, 10 kernels of size 150 were applied to the input to capture the local patterns by sliding across the input. After each 1D-CNN, a batch normalisation layer was used to normalise the output so that each layer received input with the same mean and variance. This was followed by a max-pooling layer, which selected the maximum value from each feature map window to decrease the dimensionality (Figure 6).In the last section of the DL architecture, the output of the 1D-CNN was flattened to a one-dimensional vector processed by three hidden dense layers with the ReLU activation function (Figure 6).The final classification was performed by the last DNN implementing the softmax activation function, $f_{softmax}(z_{i})=\frac{e^{z_{i}}}{\sum_{b=1}^{B}e^{z_{b}}}$, where $z_{i}$ represents the output of the last layer and B is the number of classes (i.e., breeds).

**Figure 6 ijms-26-07961-f006:**
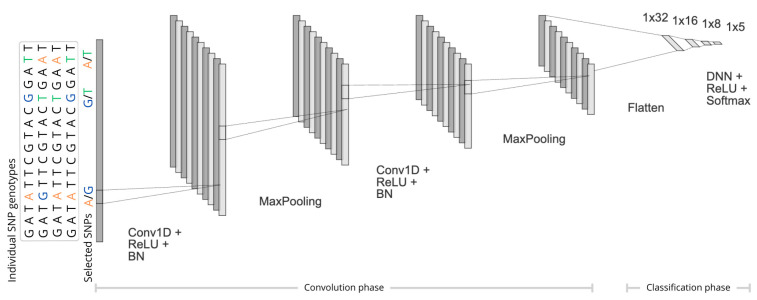
The architecture of the deep learning-based classifier. Individual SNP genotypes originate from three feature selection methods: *SNP tagging*, *1D-SRA*, or *MD-SRA*.

The Adam optimiser [31] implements the stochastic gradient descent algorithm with a default learning rate of $5.0\cdot 10^{-5}$ was applied to minimise the categorical focal-cross-entropy loss (FL) function given by the following:(4)$$FL_{i}=\alpha_{b}\times {(1-p}_{ib}{)}^{2}\times CE_{i},$$
where $\alpha_{b}=\frac{N}{\left(B\times g_{b}\right)}$ was used as suggested by Li et al. [32], with *N* representing the number of individuals, B=5 representing the number of classes and $g_{b}$ being the number of individuals assigned to class *b* in the training dataset. FL was implemented to mitigate the class frequency imbalance during optimisation by penalising the impact of more numerous classes. The SHAP values were estimated to evaluate the importance of each SNP for classification. Because of the large number of SNPs, the SHAP values were approximated by GradientExplainer [33] using nsamples = 50. The SHAP value for the p-th SNP was expressed as ${SHAP}_{p}=\frac{1}{N}\sum_{i=1}^{N}\underset{B}{\max}{(SHAP}_{i})$, where $\underset{B}{\max}{(SHAP}_{i})$ represents the maximum ${SHAP}_{i}$ across B breeds.

To evaluate the classification performance of the DL-based classifier during the training step, stratified 5-fold cross-validation was applied by randomly selecting 80% of the training individuals for training the model and leaving out the remaining 20% of the individuals for the validation dataset. The test dataset was used to assess the final performance of the trained model. The quality of classification was quantified using the Macro F-1 Score metric (Equation (5)) as follows:(5)$$F1=2\times \frac{Precision\times Recall}{Precision+Recall},$$
where $Precision=\frac{1}{B}\sum_{b=1}^{B}\frac{TP_{b}}{TP_{b}+FP_{b}}$ and $Recall=\frac{1}{B}\sum_{b=1}^{B}\frac{TP_{b}}{TP_{b}+FN_{b}}$, the $TP_{b}$ (true positive) class represents the individuals correctly classified to breed b, the $FN_{b}$ (false negative) class represents the individuals of breed b incorrectly classified to another breed, and the $FP_{b}$ (false positive) class represents individuals from another breed incorrectly classified as breed b. Additionally, the Area Under the Curve (AUC) resulting from averaging the AUCs for each individual corresponding to the Receiver Operating Characteristic (ROC) curve was constructed separately for the classification of each breed. The models were trained and evaluated on an NVIDIA Tesla P100 GPU with 328 GB RAM.

## 5. Conclusions

With the fast-growing dimensionality of datasets, which includes not only an increasing number of data records, but also an increasing number of features, the storage and computational burden is an increasing challenge [4,5]. There is a clear need for effective and efficient computing [34]. *MD-SRA* provides a very good balance between classification quality, computational intensity, and required hardware resources among the three methodologies compared. It demonstrated a classification performance that was very similar to that of 1D-SRA and markedly better than SNP tagging. Moreover, unlike *SNP tagging*, this is a universal approach that is not limited to genomic data.

## Figures and Tables

**Figure 1 ijms-26-07961-f001:**
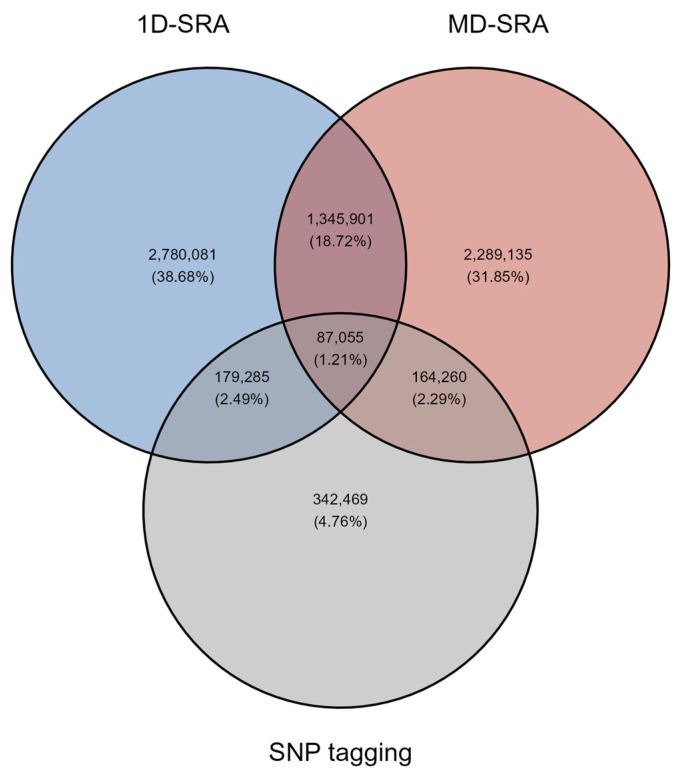
The overlap of relevant SNPs selected by the three feature selection approaches.

**Figure 2 ijms-26-07961-f002:**
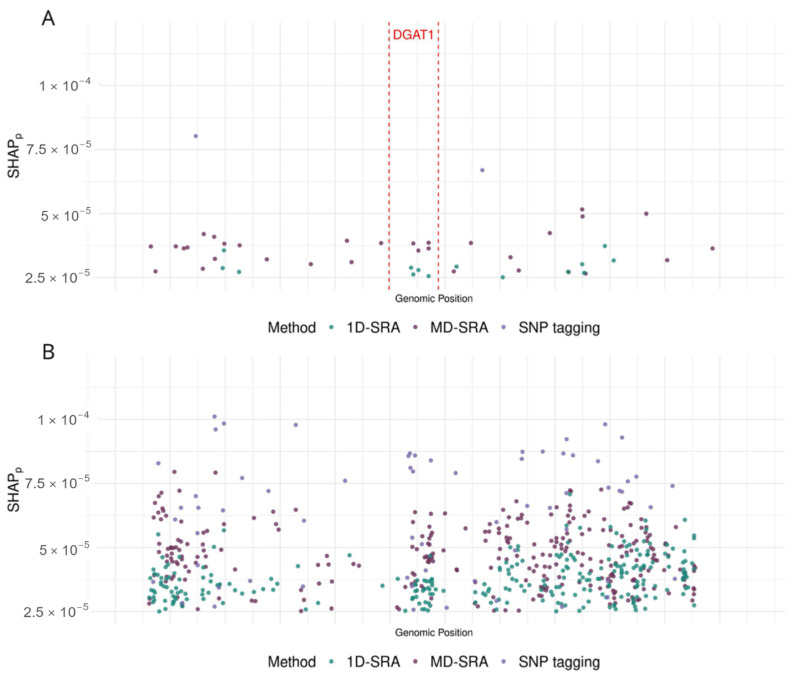
SHAP values corresponding to (**A**) the 100 kbp region (14:554,070–14:662,687) that harbours the DGAT1 gene and (**B**) the 100 kbp region (14:80,000,175–14:80,099,518) are located in a noncoding region on the distal part of the chromosome.

**Figure 3 ijms-26-07961-f003:**
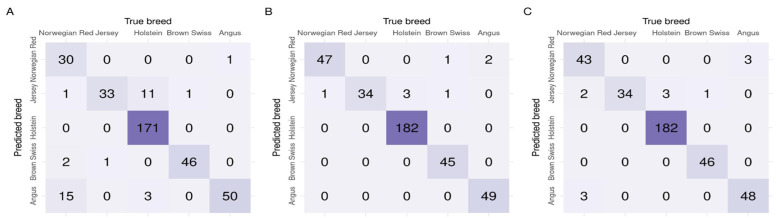
Confusion matrices of classification for the test dataset by the three feature selection algorithms (**A**) *SNP tagging*, (**B**) *1D-SRA*, and (**C**) *MD-SRA*.

**Figure 4 ijms-26-07961-f004:**
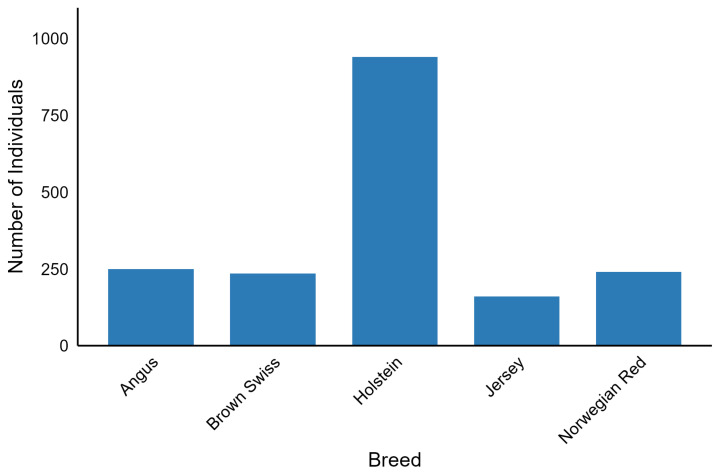
Number of selected individuals across the five breeds used in downstream analysis.

**Table 1 ijms-26-07961-t001:** Summary results of the three feature selection approaches.

	Number of Selected SNPs	Reduction Rate	Relative Computational Time	Disc Storage
*SNP tagging*	773,069	93.51%	x1.0	No intermediate files
*1D-SRA*	4,392,322	63.14%	x37.7	3.1 TB
*MD-SRA*	3,886,351	67.39%	x2.2	227 MB

**Table 2 ijms-26-07961-t002:** Macro F1-score and AUC for validation of the training datasets.

	Validation of the Training Dataset		Test Dataset	
	Macro F1-Score	AUC	Macro F1-Score	AUC
*SNP tagging*	74.65% ± 16.63%	0.9627 ± 0.0357	86.87%	0.9847
*1D-SRA*	90.09% ± 5.49%	0.9915 ± 0.0058	96.81%	0.9968
*MD-SRA*	88.22% ± 9.53%	0.9842 ± 0.0168	95.12%	0.9976

## Data Availability

The VCF files corresponding to Whole-Genome Sequence data analysed in this study are part of the 1000 Bull Genomes Run 9.0. Most of the data is publicly available in the BioProject database (https://www.ncbi.nlm.nih.gov/bioproject/ accessed on 1 September 2024)) at accession numbers PRJNA238491, PRJEB9343, PRJNA176557, PRJEB18113, PRNJA343262, PRJNA324822, PRJNA324270, PRJNA277147, PRJEB5462, and PRJNA431934. The source code for feature selection with 1D-SRA and MD-SRA algorithms and DL-based classification was uploaded on GitHub (https://github.com/ThetaUP/UHD-SRA; accessed on 30 August 2024).

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
