# Peer review of "Feature Selection Strategies for Deep Learning-Based Classification in Ultra-High-Dimensional Genomic Data"

_ijms, 2025, doi:10.3390/ijms26167961_

Round 1

Reviewer 1 Report

Comments and Suggestions for Authors

This study addresses the need for efficient computational and statistical tools for feature selection in high-dimensional genomic data. The authors classified 1,825 cows into five breeds based on 11,915,233 single nucleotide polymorphisms using SNP tagging and supervised rank aggregation approaches as well as univariate and multivariate feature clustering. The animals were then classified into breeds using a deep learning classifier consisting of convolutional neural networks. As a result, the authors showed that the proposed MD-SRA is suitable for classifying highly high-dimensional data. Due to the rapidly growing dimensionality of datasets, which include not only an increasing number of data records but also an increasing number of features, the load on storage and computational resources is becoming increasingly challenging, i.e., there is an obvious need for efficient and effective computing. The article is written in clear scientific language, including mathematical formulas and their descriptions, and is also illustrated in detail with diagrams, charts and tables. Another big plus is that the authors have posted the code on GitHub. I have no comments related to the design of the bioinformatics algorithm and the formatting of the manuscript.

Author Response

Thank you for the review of the study, we appreciate your understanding of the need of dimensionality reduction and feature selection research applied to modern genomic studies. The introduction was revised.

Reviewer 2 Report

Comments and Suggestions for Authors

Clearly define the research gap and cite recent reviews on genomic feature selection and deep learning

The research design is limited by the absence of controls for confounder

you need to annotate top SNPs with known breed-associated genes

Author Response

Comment 1: Clearly define the research gap and cite recent reviews on genomic feature selection and deep learning

Response 1: The research gap was formulated in a more clear way. Also, we added recent literature reviews:

Sartori, F.; Codicè, F.; Caranzano, I.; Rollo, C.; Birolo, G.; Fariselli, P.; Pancotti, C. A Comprehensive Review of Deep Learning Applications with Multi-Omics Data in Cancer Research. Genes 2025, 16, 648, doi:10.3390/genes16060648.

Ballard, J.L.; Wang, Z.; Li, W.; Shen, L.; Long, Q. Deep Learning-Based Approaches for Multi-Omics Data Integration and Analysis. BioData Min. 2024, 17, 38, doi:10.1186/s13040-024-00391-z.

Sujithra, L.R.; Kuntha, A. Review of Classification and Feature Selection Methods for Genome‐Wide Association SNP for Breast Cancer. In Artificial Intelligence for Sustainable Applications; Umamaheswari, K., Vinoth Kumar, B., Somasundaram, S.K., Eds.; Wiley, 2023; pp. 55–78 ISBN 978-1-394-17458-4.

Comment 2: The research design is limited by the absence of controls for confounder.

Response 2: The final goal of the study was a DL based classification, while the classes were represented by breeds. Therefore, we cannot provide a formal control as if it were a clinical study. Could you please clarify which confounder do you have in mind in the context of breed classification?

Comment 3: You need to annotate top SNPs with known breed-associated genes.

Response 3: The goal of the manuscript is to propose and discuss the modified methodology of feature preselection in genomic context, while the cattle data was used as the practical illustration of this methodology. Therefore, we deliberately avoid discussing effects of particular SNPs. To our knowledge, there exist no SNPs that can unequivocally be assigned to be breed-specific.

Round 2

Reviewer 2 Report

Comments and Suggestions for Authors

I'd  like to thank authors for their reply. I Think the manuscript is suitable for publication